# *Eucalyptus globulus* Leaf Aqueous Extract Differentially Inhibits the Growth of Three Bacterial Tomato Pathogens

**DOI:** 10.3390/plants12081727

**Published:** 2023-04-21

**Authors:** Mafalda Pinto, Cristiano Soares, Tatiana Andreani, Fernanda Fidalgo, Fernando Tavares

**Affiliations:** 1GreenUPorto-Sustainable Agrifood Production Research Centre/INOV4AGRO, Biology Department, Faculty of Sciences of University of Porto, Rua do Campo Alegre s/n, 4169-007 Porto, Portugal; 2CIBIO, Centro de Investigação em Biodiversidade e Recursos Genéticos, InBIO Laboratório Associado, Campus de Vairão, Universidade do Porto, 4485-661 Vairão, Portugal; 3Biology Department, Faculty of Sciences of University of Porto, Rua do Campo Alegre s/n, 4169-007 Porto, Portugal; 4BIOPOLIS Program in Genomics, Biodiversity and Land Planning, CIBIO, Campus de Vairão, 4485-661 Vairão, Portugal

**Keywords:** phytopathology, eucalyptus, biocide, sustainable agriculture, *Pseudomonas syringae*, *Xanthomonas euvesicatoria*, *Clavibacter michiganensis*

## Abstract

As available tools for crop disease management are scarce, new, effective, and eco-friendly solutions are needed. So, this study aimed at assessing the antibacterial activity of a dried leaf *Eucalyptus globulus* Labill. aqueous extract (DLE) against *Pseudomonas syringae* pv. *tomato* (*Pst*), *Xanthomonas euvesicatoria* (*Xeu*), and *Clavibacter michiganensis michiganensis* (*Cmm*). For this, the inhibitory activity of different concentrations of DLE (0, 15, 30, 45, 60, 75, 90, 105, 120, 135, and 250 g L^−1^) was monitored against the type strains of *Pst*, *Xeu*, and *Cmm* through the obtention of their growth curves. After 48 h, results showed that the pathogen growth was strongly inhibited by DLE, with *Xeu* the most susceptible species (15 g L^−1^ MIC and IC_50_), followed by *Pst* (30 g L^−1^ MIC and IC_50_), and *Cmm* (45 and 35 g L^−1^ MIC and IC_50_, respectively). Additionally, using the resazurin assay, it was possible to verify that DLE considerably impaired cell viability by more than 86%, 85%, and 69% after *Pst*, *Xeu*, and *Cmm* were incubated with DLE concentrations equal to or higher than their MIC, respectively. However, only the treatment with DLE at 120 g L^−1^ did not induce any hypersensitive response in all pathogens when treated bacterial suspensions were infiltrated onto tobacco leaves. Overall, DLE can represent a great strategy for the prophylactic treatment of tomato-associated bacterial diseases or reduce the application of environmentally toxic approaches.

## 1. Introduction

Nowadays, to meet the increasing food demand of an ever-growing population, agri-food production needs to step up considerably using progressively fewer resources [1]. To make matters worse, climate change is imposing additional challenges for crop growth, not only by directly restricting plant growth due to abiotic stress factors, such as high temperatures, water shortages, and salinity, but also by increasing the incidence and severity of pests and diseases [2,3].

*Solanum lycopersicum* L. (the tomato plant) is one of the most valuable crops worldwide, representing more than 15% of the world’s total vegetable production [4] and playing an important role in both human nutrition and the socio-economic activities of the main producing countries. However, its production has been severely impacted by bacterial diseases, among which are bacterial speck, bacterial spot, and bacterial cancer, caused by *Pseudomonas syringae* pv. *tomato* Okabe (*Pst*), *Xanthomonas euvesicatoria* ex Doidge (*Xeu*), and *Clavibacter michiganensis* spp. *michiganensis* Smith (*Cmm*), respectively [5]. While the first two pathogens are Gram-negative bacteria and cause parenchymatic diseases, the latter is a Gram-positive bacterium that affects both parenchymatic and vascular plant tissues [5].

*Cmm* can spread through contaminated seeds, soil, plant debris, or even dripping water or pruning tools, entering plant wounds or natural openings, such as hydathodes and stomata [4]. Disease severity depends on multiple variables, including host plant age, cultivar susceptibility, *Cmm* virulence, and environmental factors, like temperature and humidity [4]. In this way, *Cmm* proliferates via xylem, interfering with water transport and causing browning and the degradation of the internal vascular tissues, which results in rapid wilt and ultimately death when tomato plants are infected at early developmental stages. In contrast, the infection of mature tomato plants usually results in the absence of symptoms or a slower wilting process that, in general, does not affect plant survival or marketable fruit quality [4].

Infected seeds are the main dissemination vehicles of *Pst* and *Xeu*, even though their associated diseases can also propagate through contaminated transplants and pruning tools [6,7]. Both bacterial diseases can produce visible symptoms on the leaves, stems, and fruits of tomato plants. While *Pst* induces the appearance of black spots surrounded by a chlorotic halo on the leaves and by a green halo on the fruits as well as necrotic spots on the stems, *Xeu* characteristically leads to necrotic spots surrounded by chlorosis in the aerial organs of tomato plants [6,7]. To aggravate, especially under a climate change-related scenario, drastic variations of environmental factors highly contribute to increasing the susceptibility of tomato plants to bacterial spot, speck, and cancer, causing much more severe symptoms and jeopardizing crop performance and marketable fruit quality and yield [8].

Although farmers have several commercial options, such as fertilisers and pesticides, to enhance plant growth or control widespread of pests, respectively, like weeds and insects, disease management strategies are currently limited. In fact, they are mainly applied to the disinfection of pruning tools and/or the elimination of infected plants and seeds, since most phytopathogens have acquired resistance to antibiotics and tolerance to copper-based bactericides, and host-plant resistance has proved to be non-durable [5,9]. Besides, in the European Union (EU), the use of antibiotics to prevent or control crop-related diseases is forbidden according to Regulation (EU) 2019/6 (https://eur-lex.europa.eu/eli/reg/2019/6/oj; accessed on 17 July 2021). Additionally, even though the application of copper-based plant protection products is allowed, the maximum amount applied is restricted to a maximum of 28 kg per ha over a period of 7 years [regulation (EU) 2018/1981; https://eur-lex.europa.eu/eli/reg_impl/2018/1981/oj; accessed on 17 July 2021], being banning its use the ultimate goal, as they are included in the list of products to be replaced in the EU [Regulation (EU) 540/2011; https://eur-lex.europa.eu/eli/reg_impl/2011/540/oj; accessed on 17 July 2021] [10]. Therefore, it is urgent to develop new and eco-friendly strategies to effectively treat infected plants and thus, improve food production and quality. This would be particularly relevant in terms of organic agriculture, where the application of natural or naturally derived products is prioritized (Regulation (EU) 2018/848; https://eur-lex.europa.eu/eli/reg/2018/848/oj; accessed on 17 July 2021].

*Eucalyptus globulus* Labill. is an Australian hardwood tree species that has a widespread distribution throughout the world not only due to its favourable characteristics for pulpwood production but also due to the beneficial properties of its leaves [11]. Due to their richness in monoterpenoids, terpenes, flavonoids, tannins, and hydroxycinnamic acids, multiple studies have been carried out to characterize the biological activities of eucalyptus leaves, which are now known to have powerful antioxidant [12], anti-inflammatory [13], antifungal [14,15], herbicidal [16], and antimicrobial [17] properties. However, most studies conducted so far have only focused on assessing the antimicrobial activity of *E. globulus* leaf essential oils [18], whose extraction procedure is technically demanding and of low yield. Moreover, most of these works studied the antimicrobial effect of *E. globulus* leaf essential oils against human pathogenic and food-borne bacteria, concluding that they present a strong antimicrobial potential against *Escherichia coli* [12,18,19,20,21], *Pseudomonas aeruginosa* [12,20], *Salmonella* sp. [20], *Staphylococcus aureus* [18,19,20,21,22], *Bacillus* sp. [18,19], *Klebsiella pneumoniae* [12,19], and *Listeria* sp. [22,23]. Accordingly, the only study exploring the antimicrobial properties of aqueous extracts of *E. globulus* leaves focus on human pathogens, leaving much to clarify regarding their antibacterial activity against phytopathogens [24]. Actually, to the best of our knowledge, only Sabir, et al. [25] has investigated the antibacterial activity of *E. globulus* leaf essential oils against a phytopathogenic bacterial species, namely *Pst*, having verified that they presented an interesting bactericidal activity against *Pst* DC3000.

In this sense, *E. globulus* leaf extracts could represent a more practical, affordable, and feasible approach to harnessing the bioactivity of eucalyptus specialized metabolites. A previous work [26] extensively characterized the phytochemical profile of aqueous extracts prepared with fresh and dried leaves of young and mature *E. globulus* trees, suggesting potential applications for these plant extracts. This study revealed that aqueous extracts prepared with dried leaves of young trees were rich in hydroxybenzoic acids, flavonoids, and organic acids, three chemical classes whose ability to disrupt bacterial membranes has already been described, thus having strong antibacterial properties [27,28,29].

The present study aims at evaluating the antimicrobial activity of aqueous extracts prepared from dried leaves of young *E. globulus* trees against three important tomato phytopathogens, *Pst*, *Xeu*, and *Cmm.* For this, the inhibitory effect of different concentrations of the aqueous eucalyptus extract was monitored on the type strains of the three bacterial species by determining their MIC, IC_50_, and bactericidal effect by addressing their viability and hypersensitive response on tobacco leaves.

## 2. Results

### 2.1. Antibacterial Activity of the Dried Leaf Extract (DLE) against Pst, Xeu, and Cmm

After exposing *Pst*, *Xeu*, and *Cmm* for 48 h to different concentrations of the dried leaf extract (DLE), the results revealed that the eucalyptus leaf aqueous extract inhibited bacterial growth in a dose-dependent manner (Figure 1). While *Xeu* growth was inhibited by the lowest DLE concentration tested (15 g L^−1^), slightly higher doses were required to suppress *Pst* and *Cmm* growth (30 and 45 g L^−1^, respectively) (Figure 1; Table 1). For *Pst*, the half-maximal inhibitory concentration (IC_50_) of DLE was very close to the minimal inhibitory concentration (MIC), but for *Cmm* the IC_50_ was lower than that (35 g L^−1^) (Table 1). On the other hand, it was not possible to calculate the IC_50_ for *Xeu* since the lowest DLE concentration tested (15 g L^−1^) inhibited microbial growth by more than 85%. Overall, the growth data indicates that the three phytopathogenic strains present different susceptibilities to DLE, being *Xeu* the most susceptible species, followed by *Pst* and *Cmm*, the most tolerant phytopathogens.

### 2.2. Cellular Viability of Pst, Xeu, and Cmm upon 48 h of Exposure to Different DLE Concentrations

At the end of the 48 h incubation of the phytopathogens with DLE, an aliquot of each suspension from microplate wells was spread in Nutrient Agar (NA) medium to determine the minimal bactericidal concentration (MBC) for each strain. Since the exposure of the three type strains to DLE concentrations higher than their MIC, except for DLE at 250 g L^−1^, resulted in the formation of bacterial colonies indicating bacterial recovery after 48 h (data not shown), and thus not allowing the determination of the MBC, a resazurin-based assay was performed to quantify the percentage of viable cells (Figure 2). The cell viability of *Pst*, *Xeu*, and *Cmm* exposed to DLE concentrations corresponding to their MICs (30, 15, and 45 g L^−1^, respectively) was drastically reduced by 97%, 99%, and 69%, respectively, compared to the negative control (Figure 2). For *Pst* and *Xeu*, their MICs induced the highest reduction of cell viability, affecting this parameter in a similar way to that of streptomycin (ca. 98%). In contrast, the highest inhibition of the percentage of cellular viability (approximately 88%) for *Cmm* was registered after the 48 h incubation with DLE at 90 and 105 g L^−1^, in comparison with the negative control (Figure 2).

### 2.3. Hypersensitive Response (HR) on Tobacco Leaves of the Phytopathogens Treated with DLE

To evaluate the impact of DLE on the virulence of the three phytopathogens, the HR on tobacco leaves was analysed. For that purpose, *Nicotiana tabacum* L. leaves were infiltrated with each one of the three bacterial species, after these being treated for 1 h with DLE at its respective MIC (Table 1) and with a higher concentration that significantly affected the growth and viability of the tree pathogens (120 g L^−1^; Figure 1 and Figure 2). After 48 h, the incubation of the three bacterial strains with DLE at 120 g L^−1^ did not produce any HR on tobacco leaves, presenting a very similar appearance to that of the negative control (Figure 3). In opposition, the infiltration of the three pathogens treated with the respective MIC (Table 1) induced an HR identical to that of the positive control (Figure 3).

## 3. Discussion

The limited existing tools for bacterial disease management in agricultural contexts have motivated several scientific studies to investigate effective and sustainable alternatives [30]. One of the most explored approaches consists of taking advantage of the specialized metabolism of different plants, such as *Thymus* sp., *Satureja* sp., and *Origanum* sp. to control the growth of crop-related disease causal agents (reviewed by Raveau, et al. [31]). Accordingly, the leaves of *E. globulus* are characteristically rich in terpenes, flavonoids, and phenolic acids, whose antibacterial activities have already been described [28,32,33], and for that reason, eucalyptus leaf extracts can constitute a great antimicrobial strategy. As a matter of fact, a previous study has reported strong inhibitory properties of *E. globulus* extracts against human-pathogen bacteria, such as *P. aeruginosa* [24]. However, the antibacterial potential of eucalyptus extracts against phytopathogens remains unclear, since, as far as we know, only the antibacterial activity of eucalyptus essential oils has been tested against *Pst* DC3000 [25]. Based on their data, eucalyptus essential oils showed bactericidal properties, thus presenting a great potential to become a new biobactericide. In this way, this study aims to assess, for the first time, the antimicrobial properties of a eucalyptus leaf aqueous extract against three major phytopathogens causing economically important diseases in tomato plants—bacterial speck, spot, and cancer caused by *Pst*, *Xeu*, and *Cmm*, respectively. For this purpose, the impact of the exposure of the type strains of these bacterial species to different concentrations of the dried leaf eucalyptus extract was assessed, by monitoring their growth, quantifying their cellular viability, and analysing their HR on *N. tabacum* leaves.

The growth of *Pst*, *Xeu*, and *Cmm* was differentially inhibited by DLE, resulting in distinct susceptibilities to the plant extract. A recent report from our group, which analysed the metabolomic profile of aqueous extracts of eucalyptus leaves, revealed that DLE—the one prepared with dried leaves of young *E. globulus* trees –, in comparison with those prepared with fresh leaves of juvenile and mature trees and dried leaf extracts of mature trees, presented appreciable amounts of phenolic acids, more specifically, gallic acid, digalloylglucose I, IV, and V, and galloylglucose I and II [26]. Due to their great affinity for lipids, phenolic acids, a class of specialized metabolites with notorious antibacterial properties, can easily cross membranes, causing cytoplasmic acidification and consequently inducing membrane disruption [32]. For this reason, gram-negative bacteria are usually more susceptible to phenolic acids than gram-positive bacteria, thus correlating the outcomes of this study, in which *Pst* and *Xeu*, both gram-negative bacteria, were the most sensitive phytopathogens to DLE. In addition, this extract had rich contents in protocatechuic acid, a benzoic acid that had been reported to inhibit the growth of *Pseudomonas* spp., besides *Listeria monocytogenes*, *Mannheimia haemolytica*, *Pasteurella multocida*, *Escherichia coli*, and strains of *Salmonella* spp. [32] and *Staphylococcus* strains mainly by inducing membrane lysis [34]. Accordingly, Pereira, Dias, Vasconcelos, Rosa, and Saavedra [24] found that the growth of *P. aeruginosa* isolates was inhibited by an aqueous extract prepared with *E. globulus* leaves, which registered lower MIC values than those observed for *Pst* in this study. Similar results were obtained by Morales-Ubaldo, et al. [35], which reported that *Xanthomonas campestris* presented a greater susceptibility to hydroethanolic extracts of *Larrea tridentata* aerial parts in relation to *P. syringae* and *Cmm*. Yet, these two species presented identical MIC values [35]. In contrast, *X. campestris* and *Cmm* were the most susceptible species to methanolic and ethanolic extracts of *Artemisia nilagirica* leaves, with *Pst* being the most tolerant species with the highest MIC values [36]. In addition, the study of Körpe, et al. [37], which explored the antibacterial properties of aqueous and methanolic extracts of *Urtica dioica* and *Urtica pilulifera* seeds, leaves, and roots against multiple food-borne and phytopathogenic bacteria, including *Cmm*, *Pst*, and *Xeu*, concluded that *Cmm* was more affected by *U. dioica* seed and *U. pilulifera* leaf methanolic extracts than *Xeu* and *Pst.* Additionally, the MICs of both extracts for *Cmm*, *Pst*, and *Xeu* were lower than those registered in the present work for DLE. When assessing the antibacterial activity of *Allium sativa* and *Ficus carica* extracts against *Pst*, *Xeu*, and *Cmm*, Balestra, Heydari, Ceccarelli, Ovidi, and Quattrucci [5] found that *Pst* presented a higher susceptibility to these extracts than *Xeu* and *Cmm* in both in vitro and *in planta* experiments. 

This outcome diversity can be the result of multiple factors, including the type of solvent and plant species selected to obtain the botanical extracts. Indeed, the extraction of plant biomass with solvents of different polarities results in the recovery of specialized metabolites with distinct chemical properties and, thus, bioactivities [37]. Furthermore, the type of plant species selected and even the developmental stage can exert a great influence on the resultant antimicrobial activity since the pool of phytochemicals retrieved, each with distinct modes of action, will also vary. 

The MIC of DLE for *Pst*, *Xeu*, and *Cmm* induced a remarkable reduction of the percentage of cellular viability but did not interfere with the virulence of the few that remained viable, as shown by the similar intensity of the hypersensitive response of MIC infiltration on tobacco leaves for the three types of strains in comparison with the positive control. However, the treatment of bacterial species for 1 h with higher concentrations of DLE, such as 120 g L^−1^, did not induce any HR on the leaves of *N. tabacum*, showing that, at this concentration, DLE highly compromised their virulence. In addition, as hypothesized by Mariz-Ponte, et al. [38] for the treatment of *Pseudomonas syringae* pv. *actinidae* with antimicrobial peptides (AMPs), the absence of HR after the infiltration of bacteria treated with DLE at 120 g L^−1^ can also be related to the reduction of cell density. In fact, despite the different incubation periods of pathogens with DLE at 120 g L^−1^ employed in the determination of the percentage of cellular viability and in HR analysis, the 48 h incubation of the bacteria with this DLE concentration induced a large reduction in the percentage of cellular viability, suggesting that it can also have contributed to the absence of infection symptoms on tobacco leaves. 

Despite this potent inhibition of bacterial growth, the MBC for each phytopathogen was not possible to ascertain, as colony formation was always recorded. In agreement, Pereira, Dias, Vasconcelos, Rosa, and Saavedra [24] were also unable to determine the MBC for aqueous and organic solvent extracts of *E. globulus* leaves or for eucalyptus essential oils when they evaluated the antimicrobial potential of these extracts and essential oils against respiratory tract *P. aeruginosa* isolates. Still, the great reductions in cell viability induced by DLE concentrations equal to or greater than the MIC for each phytopathogen can possibly indicate that the inhibitory effects of DLE could be useful in the treatment of bacterial spot, speck, and cancer. Furthermore, the eco-friendly and low-cost nature of DLE also makes it an excellent candidate for pruning tools’ sanitizing, as suggested by Sabir, El Khalfi, Errachidi, Chemsi, Serrano, and Soukri [25] for eucalyptus leaf essential oils, as well as to reduce the application of other innovative but expensive and pollutingt treatments, as is the case with AMPs and organic solvent-based plant extracts, respectively [36,37,38]. 

In comparison with other approaches, like AMP or other single-molecule therapies, the acquisition of bacterial resistance to plant extracts, as in the case of DLE, is less likely to occur, as they comprise a complex mixture of different compounds with distinct modes of action. In fact, as mentioned earlier, DLE is composed of many phenolic acids like gallic acid, galloylglucose digalloylglucose and the benzoic acid protocatechuic acid, which induce membrane disruption, organic acids such as arabinaric and tartaric acids, and flavonoids like quercetin [26]. Organic acids have the ability to cross membranes and can disrupt ATP production and induce cytoplasmic acidification [29], whereas flavonoids display multiple modes of action in bacterial cells, as they can inhibit biofilm formation, bacterial virulence, and nucleic acid, cell envelope, and peptidoglycan synthesis, besides disrupting membrane permeability [28]. In addition, flavonoids have the ability to inhibit cell efflux pumps, a property that can be of great interest to reverse bacterial antibiotic resistance [28]. Besides, DLE had very high levels of amino acids like methylserine hexoside [23]. Non-proteogenic amino acids can act as defence compounds and are usually toxic in nature, since their structures can be similar to those of proteogenic amino acids and be mistakenly incorporated during protein biosynthesis and disrupt this process [39].

Furthermore, the application of DLE could be of particular interest in terms of integrated pest management as, in a recent publication from our group, it also showed a potent herbicidal activity against a model weed species, *Portulaca oleracea* L., when foliar-applied at the highest concentration (250 g L^−1^) [11]. So, to attain a more sustainable agronomic production, DLE could be applied in low doses to treat tomato plants-associated bacterial diseases, and in the maximum concentration to control weed proliferation. However, before its implementation, the environmental safety of this strategy should be addressed, by carefully evaluating its non-target impacts on crops, soil microorganisms and invertebrates, and even human health. Ongoing studies focused on evaluating the effects of DLE application on the physiological performance of crops are being carried out.

## 4. Materials and Methods

### 4.1. Bacterial Species and Growth Conditions

The type strains *Pseudomonas syringae* pv. *tomato* DC3000, *Xanthomonas euvesicatoria* LMG 905, and *Clavibacter michiganensis michiganensis* LMG 7333 were cultured on Nutrient Agar medium (NA; Liofilchem, Téramo, Italy) at 28 °C. 

For the evaluation of the antibacterial activity of the eucalyptus extract, synchronized cultures of each bacterial species were used. For this, a pure colony of each phytopathogen was selected, individually cultured on liquid Mueller-Hinton medium (MH), a non-selective and non-differential medium widely used for antimicrobial testing (MH; Liofilchem, Téramo, Italy), and incubated at 28 °C in a shaking incubator set for 220 rpm (Orbital Shaker, Model 3500 L, VWR^®^, Radnor, PA, USA). To obtain synchronous cultures, the strains were subcultured three times when at the exponential phase. Then, the optical density at 600 nm (OD_600_) of the bacterial suspensions was adjusted to 0.1.

To analyse the hypersensitive response in tobacco leaves, a pure colony of each phytopathogen was cultured on liquid MH medium and incubated overnight, at 28 °C in a shaking incubator set for 220 rpm.

### 4.2. Preparation of Eucalyptus Aqueous Extract

To prepare the aqueous extract, dried leaves of young *E. globulus* trees were incubated for 30 min in deionised water warmed to 70 °C, according to the procedure described by Pinto, Soares, Pereira, Rodrigues, Fidalgo, and Valente [26]. The metabolomic profile of DLE (250 g L^−1^) has been previously detailed in Pinto, Soares, Pereira, Rodrigues, Fidalgo and Valente [26]. In brief, this extract presented a great richness in gallic acids and derivatives (namely, gallic acid, galloylglucose I and II, and digalloylglucose I, IV, V), fatty acids [tetrahydroxyoctadecenoic acid, galactosylgycerol or glucosylglycerol, phloionolic acid, 6,8-dihydroxy-octanoic acid, and FA hydroxy (11:2/11:2) I], benzoic acids like protocatechuic acid, organic acids such as arabinaric acid, tartaric acid or meso-tartaric acid II, 2-hydroxypropanedioic acid, 2-methylcitric acid or homocitric acid I, and erythronic acid or threonic acid I, and amino acids like methylserine hexoside, serine glucoside, and n-1-deoxy-1-fructosylalanine, and flavonoids like quercetin [26]. Before use, the eucalyptus extract was filter-sterilized using 0.2 µm nitrocellulose filters (Whatman GmbH, Dassel, Germany). 

### 4.3. Determination of the MIC and IC_50_ of the Dried Leaf Extract (DLE) 

To assess the MIC and the IC_50_ of DLE for each phytopathogen, different concentrations of DLE (0, 15, 30, 45, 60, 75, 90, 105, 120, 135, and 250 g L^−1^) were prepared by diluting in liquid MH medium (22 g L^−1^) a stock solution, obtained by dissolving 0.22 g of MH in 10 mL of DLE (250 g L^−1^). In sterile 96-well microplates, 10 µL of the synchronized bacterial suspensions (three different suspensions for each pathogen) with an OD_600_ adjusted to 0.1 were added to 140 µL of MH medium, and each DLE concentration. Streptomycin (50 µg mL^−1^) served as an antimicrobial control in this experiment. Microplates were incubated in Multiskan™ GO (Thermo Fisher Scientific, Waltham, MA, USA) with constant shaking for 48 h at 28 °C and hourly readings of the OD_600_.

### 4.4. Assessment of the Cell Viability upon Exposure to DLE

The viability of the bacterial cells incubated for 48 h with different concentrations of DLE and streptomycin was evaluated using the resazurin method, essentially as described by Lee and Jain [40]. Briefly, when viable cells come into contact with resazurin (bluish with absorbance at 570 nm), they metabolize it and produce a pink product, the resorufin (pink with absorbance at 600 nm). By quantifying the absorbance at 570 (Abs_570nm_) and 600 nm (Abs_600nm_) at the beginning end of the incubation period (t_0_ and t_4_, respectively), the percentage of viable bacterial cells can be estimated. So, at the end of the 48 h incubation of each microplate, the content of each well was centrifuged for 5 min at 5800× *g*, to eliminate the interference imposed by DLE colour in absorbance readings. After carefully discarding the supernatant, the pellet was resuspended in sterile phosphate buffer saline (PBS; 8 g L^−1^ NaCl; 0.2 g L^−1^ KCl; 2.9 g L^−1^ Na_2_HPO_4_·12H_2_O; 0.2 g L^−1^ KH_2_PO_4_; pH 7.0) for the same final volume (150 µL). The suspensions were distributed in new microplates, to which 10 µL of 1 mM resazurin solution (resazurin sodium salt; Sigma-Aldrich, St. Louis, MO, USA) prepared in PBS buffer were added. As control situations, wells containing 160 µL of sterile PBS and 150 µL of PBS to which 10 µL of the resazurin solution were added were both considered. The microplates were incubated for 4 h at 28 °C with constant agitation in the microplate reader Multiskan™ GO and the Abs_570nm_ and Abs_600nm_ were read at t_0_ and t_4_. The percentage of viable cells was calculated according to the following formula [40]:
$$\begin{array}{l} \mathrm{Percentage}\;\mathrm{of}\;\mathrm{viable}\;\mathrm{cells}\\ \;\frac{\varepsilon_{\mathrm{resazurin}\_600\mathrm{nm}}\times {\mathrm{Abs}}_{570{\mathrm{nm}}_{t4}-}\varepsilon_{\mathrm{resazurin}\_570\mathrm{nm}}\times {\mathrm{Abs}}_{600{\mathrm{nm}}_{t4}}}{\varepsilon_{\mathrm{resorufin}\_600\mathrm{nm}}\times {\mathrm{Abs}}_{600{\mathrm{nm}}_{t0}-}\varepsilon_{\mathrm{resorufin}\_570\mathrm{nm}}\times {\mathrm{Abs}}_{570{\mathrm{nm}}_{t0}}} \end{array}$$
where

ε_resazurin_600nm_ and ε_resazurin_570nm_ is the extinction molar coefficient of resazurin at absorbances of 600 and 570 nm_;_

ε_resorufin_600nm_ and ε_resorufin_570nm_ is the extinction molar coefficient of resorufin at absorbances of 600 and 570 nm;

Abs_570nm_t0_ and Abs_570nm_t4_ represents the absorbance at 570 nm at the beginning and at the end of the incubation period, respectively;

Abs_600nm_t0_ and Abs_600nm_t4_ represents the absorbance at 600 nm at the beginning and at the end of the incubation period, respectively.

### 4.5. Analysis of the HR in Tobacco Leaves 

Although tomato plants are the primary hosts of the studied phytopathogen species, tobacco plants were used as a model species to assess the effect of DLE on pathogens’ virulence. So, after sowing tobacco seeds (*Nicotiana tabacum* cv. havana) in peat substrate (Siro, Mira, Portugal), seedlings were transferred to plastic pots containing 600 mL of this substrate. Before infiltrating the different treatments in their leaves, plants grew for a month in a growth chamber at 23 °C under a photoperiod of 16 h/8 h light/dark, 50% relative humidity, and photosynthetically active radiation (PAR) of 120 mmol m^−2^ s^−1^, where the hypersensitive response assay was carried out. 

The HR of tobacco leaves exposed to each phytopathogen treated with the corresponding MIC, shown in Table 1, and a concentration that presented antimicrobial activity against all tested bacterial species (120 g L^−1^ DLE) were analysed following the protocol of Mariz-Ponte, Regalado, Gimranov, Tassi, Moura, Gomes, Tavares, Santos, and Teixeira [38]. The incubation period of bacterial cells in different situations was optimized in the preliminary assays. Briefly, before infiltration on tobacco leaves, bacterial suspensions were incubated with the selected situations for 1, 2, 4, 6, 8, and 16 h, using *Pst* as a model (data not shown). Results revealed that all incubation periods produced similar results, and for that reason, we selected the minimum incubation period (1 h) to perform the final assay (data not shown). 

A pure colony of each phytopathogen was cultured on liquid MH medium and incubated overnight at 28 °C in a shaking incubator set for 220 rpm. Afterwards, the bacterial suspensions were centrifuged at 2500 rpm (Thermo Scientific centrifuge bucket 75003655, Mega Star 1.6 R, VWR^®^, Radnor, PA, USA) for 5 min at room temperature. After removing the supernatant, the pellet was resuspended in PBS until the OD_600_ reached 0.2. Then, each bacterial suspension was added in a 1:1 proportion to PBS, DLE at twice the concentration of the MIC of each phytopathogen (30 g L^−1^ for *Xeu*, 60 g L^−1^ for *Pst*, and 90 g L^−1^ for *Cmm*), and 240 g L^−1^ DLE. These concentrations of DLE were prepared by diluting DLE in PBS. Sterile PBS was used as a negative control. The suspensions were incubated at room temperature in a shaking incubator set to 115 rpm (Orbi-Shaker, Benchmark, Sayreville, NJ, USA) for 1 h. To eliminate the possible interference of DLE in the HR, suspensions were centrifuged at 2500 rpm (Thermo Scientific centrifuge bucket 75003655, Mega Star 1.6 R, VWR^®^, Radnor, PA, USA) for 5 min at room temperature, and the pellet was resuspended in PBS buffer for the same final volume. Finally, 1 mL of the following situations was infiltrated on three different leaves of tobacco plants containing two pairs of fully expanded leaves: Negative control—sterile PBS;Positive control—bacterial suspension of *Pst*, *Xeu*, or *Cmm* on PBS;MIC—bacterial suspension of *Pst*, *Xeu*, or *Cmm* treated with the concentration of DLE corresponding to their MIC (determined in Section 4.3.—30 g L^−1^ for *Pst*, 15 g L^−1^ for *Xeu*, and 45 g L^−1^ for *Cmm*)*;*120 g L^−1^ DLE—bacterial suspension of *Pst*, *Xeu*, or *Cmm* treated with DLE at 120 g L^−1^.

The HR was photographically recorded 24 h and 48 h post infiltration. This experiment was performed three times for each phytopathogen.

### 4.6. Statistical Analysis

The results were expressed as mean ± standard deviation (SD) and all experimental situations comprised at least three experimental replicates (*n* ≥ 3). The homogeneity and normality of the data were verified using the Brown-Forsythe and Shapiro-Wilk tests, respectively. Whenever needed, data were transformed to meet the analysis of variance (ANOVA) assumptions. To determine the MIC for each phytopathogen, a one-way ANOVA, followed by the Tukey *post-hoc* test, was applied to the values corresponding to the end of the exponential growth phase (28 h for *Pst*, 34 h for *Xeu*, and 47 h for *Cmm*). The MIC corresponded to the lowest concentration of DLE that was capable of significantly inhibiting bacterial growth (*p* ≤ 0.05). In turn, the IC_50_ for each phytopathogen, i.e., the concentration of DLE that caused a decline of 50% of the bacterial population, was calculated by linear regression, using the values of growth corresponding to the end of the exponential growth phase. To assess the effect of different concentrations of DLE on the percentage of viable cells for each pathogen species, a one-way ANOVA followed by the Tukey *post-hoc* test was used, considering statistically significant differences at *p ≤* 0.05. Statistical analyses were performed using GraphPad Prism^®^ 7.0 software (GraphPad Software, San Diego, CA, USA).

## 5. Conclusions

This study characterized for the first time the antibacterial activity of an aqueous extract of *E. globulus* leaves against phytopathogenic species, namely, the ones affecting tomato production—*Pst*, *Xeu*, and *Cmm*. Dried eucalyptus leaf extracts presented bacteriostatic activity, severely inhibiting the growth and the cellular viability of the type strains of these pathogens, which showed different susceptibilities to the extract—*Cmm* was the most tolerant species, followed by *Pst* and *Xeu*, which was the most susceptible species. Additionally, the treatment of these phytopathogen strains with a high concentration of DLE inhibited the hypersensitive response on *N. tabacum* leaves. For these reasons, DLE can have a great potential to be used in the prophylactic treatment of tomato-associated diseases, in the disinfection of pruning tools, or even to complement other effective but expensive or environmentally toxic approaches, allowing the reduction of their application.

## Figures and Tables

**Figure 1 plants-12-01727-f001:**
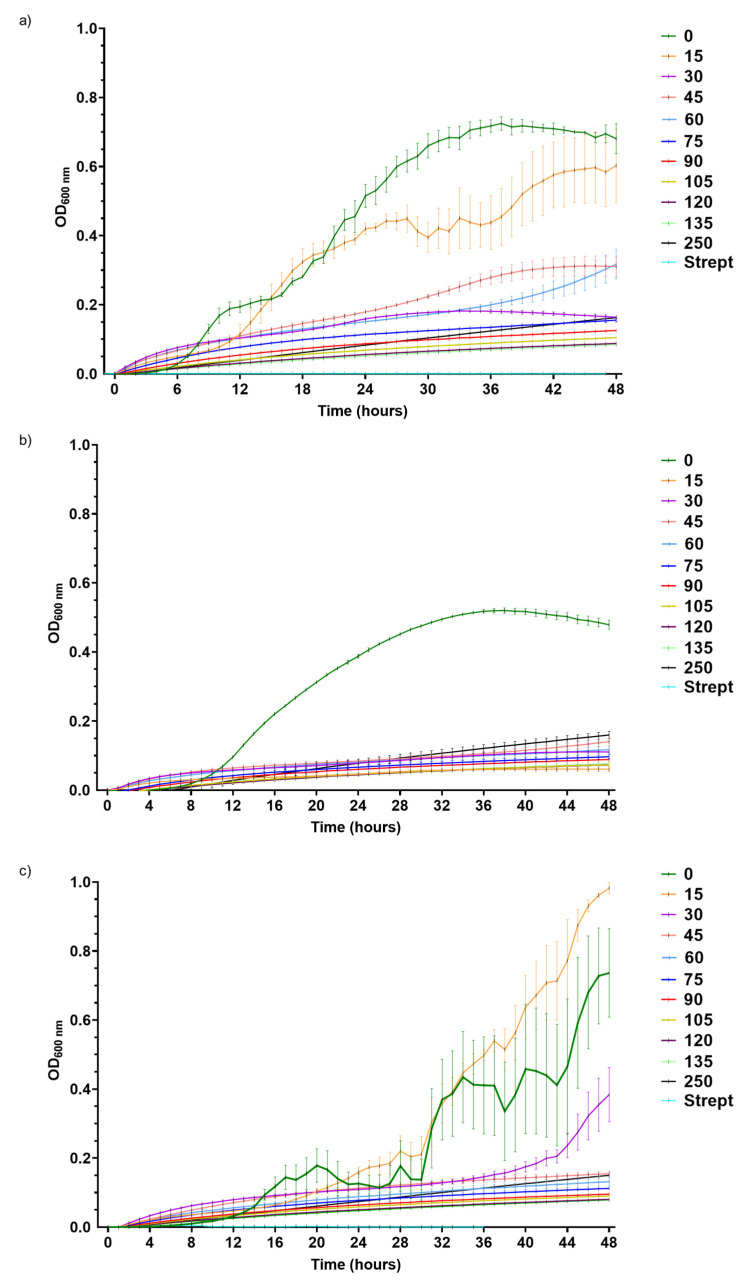
Growth curves of *Pseudomonas syringae* pv. *tomato* DC3000 (**a**), *Xanthomonas euvesicatoria* LMG 905 (**b**), and *Clavibacter michiganensis michiganensis* LMG 7333 (**c**) incubated for 48 h in different concentrations of the dried eucalyptus leaf extract (0, 15, 30, 45, 60, 75, 90, 105, 120, 135, and 250 g L^−1^) prepared in MH medium. Streptomycin (Strept) was used as a control of growth inhibition (50 µg mL^−1^). Each value indicates the average of three independent determinations.

**Figure 2 plants-12-01727-f002:**
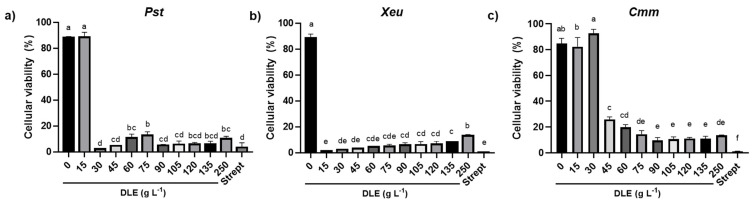
The percentage of viable cells was determined by the resazurin assay of *Pseudomonas syringae* pv. *tomato* DC3000 (**a**), *Xanthomonas euvesicatoria* LMG 905 (**b**), and *Clavibacter michiganensis michiganensis* LMG 7333 (**c**) incubated for 48 h with different concentrations of the dried eucalyptus leaf extract (0, 15, 30, 45, 60, 75, 90, 105, 120, 135, and and 250 g L^−1^) prepared in MH medium. Streptomycin (Strept) was used as a control for growth inhibition (50 µg mL^−1^). Different letters above the bars denote significant differences (*p* ≤ 0.05), according to Tukey’s *post-hoc* test.

**Figure 3 plants-12-01727-f003:**
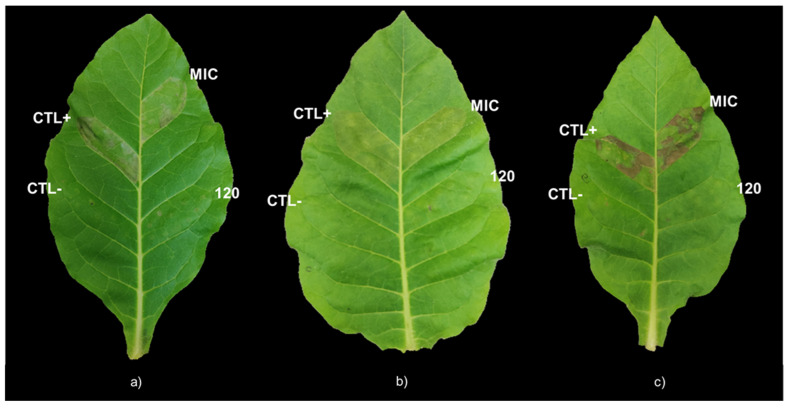
Hypersensitive response on tobacco leaves upon 48 h of infiltration of PBS buffer (negative control; CTL−), and of *Pseudomonas syringae* pv. *tomato* DC3000 (**a**), *Xanthomonas euvesicatoria* LMG 905 (**b**), and *Clavibacter michiganensis michiganensis* LMG 7333 (**c**) resuspended in PBS buffer (positive control; CTL+) and treated with the minimal inhibitory concentration (MIC), and DLE at 120 g L^−1^ (120).

**Table 1 plants-12-01727-t001:** Minimal inhibitory concentration (MIC) and half-inhibitory concentration (IC_50_) of the dried eucalyptus leaf extract for *Pseudomonas syringae* pv. *tomato* DC3000 (*Pst*), *Xanthomonas euvesicatoria* LMG 905 (*Xeu*), and *Clavibacter michiganensis michiganensis* LMG 7333 (*Cmm*).

	MIC	IC_50_
*Pst*	30 g L^−1^	29 g L^−1^
*Xeu*	15 g L^−1^	<15 g L^−1^
*Cmm*	45 g L^−1^	35 g L^−1^

## Data Availability

Not applicable.

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
