# Peer review of "Eucalyptus globulus Leaf Aqueous Extract Differentially Inhibits the Growth of Three Bacterial Tomato Pathogens"

_plants, 2023, doi:10.3390/plants12081727_

Round 1

Reviewer 1 Report

The logical structure of the manuscript is flawed: First, the authors dwell on the need for biopesticides alternative to synthetic pesticides to control phytopathogenic bacteria that attack tomato plants. Then, they propose that a eucalypt leaf extract (DLE) could help to treat three bacterial phytopathogens. Afterward, they demonstrated that the DLE differentially inhibited three strains of phytopathogens in vitro, but they ignored the well-documented variability among strains of the same phytopathogen species in virulence and susceptibility to antimicrobials. Then, they test the ability of damaged bacteria to induce HR in tobacco plants, but there was no explanation for the use of tobacco instead of Tomato plants. Also, instead of testing the DLE with diseased plants, they focused on the hypersensitive response (HR). The induction of the HR test with bacteria exposed to DLE has no relationship with controlling the infection process or disease development in tomato plants.

Furthermore, they affirm that “Still, the great reductions on cell viability induced by DLE concentrations equal to or greater than the MIC for each phytopathogen clearly show that the inhibitory effects of DLE could be particularly useful in the treatment of bacterial spot, speck, and cancer since it can be an effective aid for tomato plants’ defences successfully combat infections.” That affirmation is baseless, because an in vitro test with one strain of each bacteria is insufficient to suggest that an extract “could be particularly useful in the treatment of bacterial….”.

“In addition, the use of DLE may constitute a feasible strategy for prophylactic treatments of tomato-associated diseases, like the ones induced by the type strains of this study or others such as Ralstonia solanacearum, which causes devastating economic impacts on tomato production [36].” Again, that speculation is farfetched.

Not surprisingly, the dozens of secondary metabolites in the eucalyptus extract have activity against the tested bacteria and can attack many metabolic targets. Also, all plants have secondary metabolites that function as defenses against attacking organisms. Therefore, the conclusion that DLE extract has bactericidal or bacteriostatic activity is only relevant if the extract functions as a biopesticide when applied to living plants. It would also be significant as a disinfectant if the plant extract does not degrade or lose efficacy when stored and distributed for use and is innocuous to humans. Thus, speculation about the potential of DLE extract is premature and unwarranted if based on the in vitro experiments performed in this study.

 Additional points:

- The virulence of the phytopathogens should be evaluated with bacteria unexposed to the leaf extract because the DLE can kill or damage bacterial cells. The hypersensitive response (HR) depends on elicitors derived from the attacked cells or bacteria enzymes or fragments; thus, which molecules elicited the HR is uncertain.

- The authors should provide evidence of the tomato plant´s defense mechanisms against bacteria phytopathogens.

- The visual evaluation of the HR is insufficient because a quantitative approach is missing, for example, measuring phytoalexins or other secondary metabolites involved in the HR.

- Proving antimicrobial activity in vitro from a single extract is only the first step in a very long way toward an agronomically effective, economical, and environmentally viable biopesticide. Thus, to claim that the found antimicrobial activity is promising for a biopesticide is farfetched and groundless.

Author Response

We acknowledge the reviewer for the analysis of our manuscript. We have addressed his/her valuable comments and corrections, and all alterations have been highlighted in the text of the revised MS.

[Reviewer comments (R) and authors’ answers (A)]

(R) The logical structure of the manuscript is flawed: First, the authors dwell on the need for biopesticides alternative to synthetic pesticides to control phytopathogenic bacteria that attack tomato plants. Then, they propose that a eucalypt leaf extract (DLE) could help to treat three bacterial phytopathogens. Afterward, they demonstrated that the DLE differentially inhibited three strains of phytopathogens in vitro, but they ignored the well-documented variability among strains of the same phytopathogen species in virulence and susceptibility to antimicrobials

(A) Although we do understand the reviewer’s perspective, it is not feasible to address in these studies the full infrasubspecific diversity of a given pathogenic species. Having that in mind, we decided to use representative strains of three different species, i.e. the type strains, from each phytopathogen species in our experimental design to evaluate the antibacterial activity of DLE. Obviously it is now important to extend this analysis to numerous other strains that might represent the diversity within each species, but most importantly that represent the bacteria associated to recent plant infections. This was, if we will, a proof of concept, to set up the foundations for an upscale study.

(R) Then, they test the ability of damaged bacteria to induce HR in tobacco plants, but there was no explanation for the use of tobacco instead of Tomato plants. Also, instead of testing the DLE with diseased plants, they focused on the hypersensitive response (HR). The induction of the HR test with bacteria exposed to DLE has no relationship with controlling the infection process or disease development in tomato plants.

(A) We understand the concerns of the reviewer and for that reason, we added in the main text of the material and methods section an explanation for the use of tobacco plants instead of tomato plants. However, we would like to mention that in this study we are using the hypersensitive response, a natural response produced by plants, typically characterized by the appearance of necrotic spots or halos to prevent the dissemination of bacterial infections, to analyse the impact that the eucalyptus extract has on the virulence of the tested phytopathogens. We have used tobacco plants since it is a widely used model to assess the impact of antimicrobial drugs on plants (doi: 10.3390/molecules26051461; doi: 10.3390/biology11111685). As mentioned above in follow-up studies, using a broader range of strains we will assess, in field trials, the antibacterial activity of the eucalyptus extract in infected tomato plants, the primary hosts of the studied bacterial species, in order to determine the practical applicability of the eucalyptus extract. In this context, we consider extremely important the quantification of viable cells in in vivo material, and we intend to determine that by counting the number of colony-forming units per mL over time, in order to understand the symptom evolution of plants upon treatment with the eucalyptus extract.

(R) Furthermore, they affirm that “Still, the great reductions on cell viability induced by DLE concentrations equal to or greater than the MIC for each phytopathogen clearly show that the inhibitory effects of DLE could be particularly useful in the treatment of bacterial spot, speck, and cancer since it can be an effective aid for tomato plants’ defences successfully combat infections.” That affirmation is baseless, because an in vitro test with one strain of each bacteria is insufficient to suggest that an extract “could be particularly useful in the treatment of bacterial….”.

(A) We understand the reviewers’ concerns and altered the main text based on her/his comments.

(R) “In addition, the use of DLE may constitute a feasible strategy for prophylactic treatments of tomato-associated diseases, like the ones induced by the type strains of this study or others such as Ralstonia solanacearum, which causes devastating economic impacts on tomato production [36].” Again, that speculation is farfetched.

(A) We agree with the reviewer since we did not test the antimicrobial activity against other bacterial species like R. solanacearum. So, we decided to erase that affirmation from the discussion section.

(R) Not surprisingly, the dozens of secondary metabolites in the eucalyptus extract have activity against the tested bacteria and can attack many metabolic targets. Also, all plants have secondary metabolites that function as defenses against attacking organisms. Therefore, the conclusion that DLE extract has bactericidal or bacteriostatic activity is only relevant if the extract functions as a biopesticide when applied to living plants. It would also be significant as a disinfectant if the plant extract does not degrade or lose efficacy when stored and distributed for use and is innocuous to humans. Thus, speculation about the potential of DLE extract is premature and unwarranted if based on the in vitro experiments performed in this study.

(A) We think that the results obtained in this study are very promising since they characterize, for the first time, the antibacterial activity of an eco-friendly extract, prepared without the use of pollutant organic solvents, against three type strains of bacterial species that cause great economic impacts on tomato production and for which no treatments are available. In fact, our results show that this extract had strong inhibitory effects against these strains, significantly impacting their viability (by more than 69%). In this way, these outcomes can pave the way for new studies exploring the antibacterial activity of aqueous extracts prepared with dried eucalyptus leaves against tomato plants infected with these bacterial species or even other species like R. solanacearum. As mentioned before, we intend to evaluate the antibacterial effects of DLE on infected tomato plants in field trials. In addition, as we suggested in our manuscript, studies on the potential use of DLE to disinfect pruning tools could also be explored in future studies.

(R) The virulence of the phytopathogens should be evaluated with bacteria unexposed to the leaf extract because the DLE can kill or damage bacterial cells. The hypersensitive response (HR) depends on elicitors derived from the attacked cells or bacteria enzymes or fragments; thus, which molecules elicited the HR is uncertain.

(A) The reviewer is absolutely right. Having that in mind, we included a control situation in our experiment, which we called positive control (CTL +), in which bacterial suspensions of Pst, Xeu, and Cmm were not treated with DLE and were just resuspended in PBS buffer. The HR from the bacterial suspensions treated with different DLE concentrations were compared with the HR from the corresponding positive control situation and also with the negative control (infiltration of PBS buffer alone in tobacco leaves).

(R) The authors should provide evidence of the tomato plant´s defense mechanisms against bacteria phytopathogens.

(A) We would like to thank the reviewer for her/his suggestion. In fact, in future studies, we intend on assessing the antibacterial potential of DLE against tomato plants infected with the type strains of Xeu, Pst, and Cmm. Focus will be given to the effect of DLE on mitigating disease symptoms and on the physiological performance of tomato plants, namely on components of the antioxidant defense system.

(R) The visual evaluation of the HR is insufficient because a quantitative approach is missing, for example, measuring phytoalexins or other secondary metabolites involved in the HR.

(A) We agree with the reviewer that quantifications can give precious information regarding biological responses and are very helpful to compare the effectiveness of different treatments. However, we believe that the HR between the distinct treatments was so clear and different from the corresponding control situations that in this particular case, a quantitative approach would result in the same conclusions.

(R) Proving antimicrobial activity in vitro from a single extract is only the first step in a very long way toward an agronomically effective, economical, and environmentally viable biopesticide. Thus, to claim that the found antimicrobial activity is promising for a biopesticide is farfetched and groundless.

(A) We agree with the reviewer. As previously commented, we think that the results of this manuscript can pave the way to several new studies exploring the potentialities and the real applicability of this strategy for crop bacterial disease management. Moreover, for DLE to be used in agriculture to treat crop bacterial diseases, a series of additional studies will have to be carried out in order to ensure its environmental safety for non-target species, especially for the soil microbial community.

Reviewer 2 Report

Dear Authors,

The Manuscript ID:  plants-2274491, Titled “Eucalyptus globulus leaf aqueous extract differentially inhibits the growth of three bacterial tomato pathogens” is well-designed. It states the purpose of the research, the principal results and conclusions.

Based on the evaluation of its originality, significance of content, scientific soundness, and interest to readers, a minor revision is suggested before the article may be considered for acceptance. Specific suggestions and comments are provided below.

The title is summarizing and informative. It declares clearly the aims of the study.

The Abstract is factual and well-structured. It states briefly about antibacterial activity of a dried leaf Eucalyptus globulus aqueous extract  against Pseudomonas syringae pv. tomato, Xanthomonas euvesicatoria, and Clavibacter michiganensis michiganensis.  Nevertheless, the used methods are not mentioned and I suggest to be written which they are.

The Introduction presents comprehensive data concerning the settled aims. The literature survey is wide-ranging. Nevertheless, I would suggest and recommend to include some missing sources, e.g.:

Awol Mekonnen, Berhanu Yitayew, Alemnesh Tesema, Solomon Taddese. In Vitro Antimicrobial Activity of Essential Oil of Thymus schimperi, Matricaria chamomilla, Eucalyptus globulus, and Rosmarinus officinalis. International Journal of Microbiology. 2016; Article ID 9545693. https://doi.org/10.1155/2016/9545693

Raho G Bachir, M Benali. Antibacterial activity of the essential oils from the leaves of Eucalyptus globulus against Escherichia coli and Staphylococcus aureus. Asian Pacific Journal of Tropical Biomedicine. 2012; 2(9): 739-742.

Manliang Tan, Ligang Zhou,Yongfu Huang,Ye Wang,Xiaojiang Hao & Jingguo Wang. Antimicrobial activity of globulol isolated from the fruits of Eucalyptus globulus Labill. Natural Product Research. Formerly Natural Product Letters. 2008; 22 (7): 569-575.

Material and methods are described in details, divided with subheadings according to the procedure. There is an adequacy of the methodology.

Results and discussion. The results are complete and clear. The discussion explores the significance of the results. The work is well-illustrated.

The Conclusions are well summarized based on the antibacterial activity of an aqueous extract of E. globulus leaves against phytopathogenic species. Future aspects in the prophylactic treatment of tomato-associated diseases and etc. are included too.

Author Response

We want to thank the reviewer for reviewing our manuscript and for the positive comments. We have carefully analysed your suggestions and implemented them in the manuscript (highlighted in the text).

Reviewer 3 Report

The study is quite interesting as it addresses a problem related to Solanum lycopersicum L. (tomato plant) which is used for human nutrition and represents income for various sectors of agriculture. It is known that the methods used to contain certain infections end up leading to phenomena of resistance, which may require the use of greater amounts of pesticides and fertilizers. Thus this study aimed to evaluate the antimicrobial activity of aqueous extracts of dry leaves of young Eucalyptus globulus trees against three phytopathogens, Pseudomonas syringae pv. tomato (Pst), Xanthomonas euvesicatoria (Xeu), and Clavibacter michiganensis michiganensis (Cmm). The inhibitory effect of different concentrations was monitored of the aqueous extract in the type strains of the three species of bacteria determining MIC, IC50 and bactericidal effect evaluating the viability as well as the hypersensitivity response in tobacco leaves.

The authors found a promising alternative to circumvent the problems mentioned, since the emergence of bacterial resistance against an extract is a much more difficult event than against pure substances.

Author Response

We are grateful to the reviewer for reviewing our manuscript. We also want to thank the reviewer for the great comments made on our manuscript.
